# Exploring Semantic Variations in GAN Latent Spaces via Matrix Factorization

**Andrey Palaev[1], Rustam A. Lukmanov[1] & Adil Khan[1,2]**
[1]Innopolis University, [2]University of Hull
`a.palaev@innopolis.university`, {`r.lukmanov, a.khan`}`@innopolis.ru`
`a.m.khan@hull.ac.uk`

## Abstract

Controlled data generation with GANs is desirable but challenging due to the non-linearity and high dimensionality of their latent spaces. In this work, we explore image manipulations learned by GANSpace, a state-of-the-art method based on PCA. Through quantitative and qualitative assessments we show: (a) GANSpace produces a wide range of high-quality image manipulations, but they can be highly entangled, limiting potential use cases; (b) Replacing PCA with ICA improves the quality and disentanglement of manipulations; (c) The quality of the generated images can be sensitive to the size of GANs, but regardless of their complexity, fundamental controlling directions can be observed in their latent spaces. The code is available at this URL: `https://github.com/Palandr1234/exploring-gan-latent-spaces`

## 1 Introduction

Generative Adversarial Networks (GANs) (Goodfellow et al., 2014) have enabled significant progress in data synthesis, including the generation of images, video, and text (Karras et al., 2018). GANs consist of two networks: a discriminator that differentiates between real and generated data, and a generator that produces realistic data from random latent vectors. However, the lack of control over the data generated by GANs limits their usage not only for images but also for other types of data. To overcome this limitation, several methods of image manipulation in the latent space of GANs have been developed such as InterFaceGAN (Shen et al., 2020) and GANanalyze (Goetschalckx et al., 2019). Our work is aimed to explore the latent space manipulations in GANs to gain insight into the underlying processes that enable the generation of new and semantically meaningful images. Specifically, we focused on a state-of-the-art (SOTA) unsupervised method - GANSpace (Härkönen et al., 2020), which utilizes Principal Component Analysis (PCA) (Pearson, 1901) to find orthogonal directions along which new semantic axes can be located. We performed both a visual and quantitative evaluation of GANSpace on StyleGAN2 (Karras et al., 2020), SOTA large-scale GAN producing realistic images, and GAN from Liu et al. (2021), light-weight GAN that requires significantly fewer resources and less training time but still produces high-quality images. Finally, we replaced PCA with Independent Component Analysis (ICA) (Hyvärinen & Oja, 2000) because ICA was found to be a preferred option as it searches for statistically independent components and allows for a more disentangled representation of the data. This work also represents our first attempt towards the goal of exploring larger questions of the semantic organization of latent spaces and their interpretability.

## 2 Background

PCA (Pearson, 1901) is a statistical technique that transforms data into a new coordinate system and maximizes variance observed in the data. However, it may not be effective when the components with the most variance do not contain the most valuable information or when the underlying components are not orthogonal, leading to highly entangled results. In contrast, ICA (Hyvärinen & Oja, 2000) is a computational approach that breaks down a multivariate signal into additive components by assuming that at most one component is Gaussian and that the components are statistically in-

dependent from each other. ICA pays attention to statistical independence, which may enable the discovery of disentangled manipulations in the latent space of GANs.

## 3 METHODS

In this work, we evaluated the transformations obtained by GANSpace (Härkönen et al., 2020) for both StyleGAN2 (Karras et al., 2020) and GAN from Liu et al. (2021) both visually and quantitatively. For StyleGAN2, as was done by Härkönen et al. (2020) we applied GANSpace in $\mathcal{W}$ latent space. For the GAN from Liu et al. (2021), we performed the approach analogical to what the authors of GANSpace did to BigGAN (Brock et al., 2019). Specifically, we sampled N random latent vectors and processed them through the $conv1$ layer of the GAN to get N feature vectors. Then, we applied PCA to these feature vectors to get the principal component matrix $V$ and obtained the matrix of transformation in the original latent space using the least squares method.

To evaluate the quality of the transformations found by GANSpace, we generated 1000 images for both GANs and applied one random transformation from a set found by GANSpace to each image. We calculated the Fréchet inception distance (FID) (Heusel et al., 2017) between the original generated images and the transformed ones to measure the similarity between the two sets of images. A lower FID score indicates higher similarity. Finally, we substituted PCA with ICA in the GANSpace method, conducted a similar evaluation of the resulting manipulations, and subsequently compared them to those obtained using the original GANSpace.

## 4 RESULTS AND DISCUSSION

The examples of manipulations obtained by GANSpace (Härkönen et al., 2020) are shown in Fig. 1 for StyleGAN2 (Karras et al., 2020) and Fig. 2 for GAN from Liu et al. (2021). GANSpace produced high-quality transformations for both GANs, as demonstrated by low FID scores (Table 1). However, we observed that PCA does not fully differentiate entangled directions in the obtained transformations, as it focuses only on maximizing the variance and orthogonality.

To address the problem of entanglement, we replaced PCA with ICA in the GANSpace processing pipeline. By replacing PCA with ICA, we found that increasing the number of components leads to a significant increase in the disentanglement of the manipulations (Figs. 3, 4). GANSpace with ICA discovers a more diverse set of manipulations than GANSpace with PCA, and the FID score improved as the number of components increased (Table 2). For the GAN from Liu et al. (2021), ICA was applied at the $conv1$ layer, and the resulting set of manipulations can be characterized as more diverse than those obtained with PCA and similar to one produced by ICA with StyleGAN2. However, the increased number of ICA components only slightly increases the disentanglement and does not change the manipulation quality (Figs. 5, 6, Table 3). Our results suggest that ICA may help achieve disentangled image manipulations in the latent space of GANs, but its effectiveness depends on the underlying disentanglement of the feature space and the careful choice of the number of components. Despite the difference in the number of parameters and dimensionality of the feature spaces, both the original GANSpace and ICA discover the same set of manipulation directions for both GANs.

More details of the results can be found in Section B.

## 5 CONCLUSION

In conclusion, this study examined image manipulation in the latent space of GANs, focusing on GANSpace applied to two GANs: StyleGAN2 and the GAN from Liu et al. (2021). GANSpace demonstrates effective image manipulations on both GAN models, however, the manipulations exhibit a high degree of entanglement. To address this limitation, ICA was tested as a replacement for PCA in GANSpace. The findings reveal that ICA can potentially produce more disentangled manipulations and a wider variety of transformations compared to PCA. However, the number of components needs to be carefully selected to achieve disentanglement, and the disentanglement of manipulations depends on whether the original latent or feature space has the underlying independent components. The quantitative evaluation using FID scores supported the visual evaluation results. Furthermore, our results suggest that the primary controlling directions within the latent space remain consistent regardless of the complexity of the investigated GAN models.

ACKNOWLEDGEMENTS

This research has been financially supported by The Analytical Center for the Government of the Russian Federation (Agreement No. 70-2021-00143 dd. 01.11.2021, IGK 000000D730321P5Q0002)

URM STATEMENT

The first author of this paper, namely Andrey Palaev, meets the URM criteria of the ICLR 2023 Tiny Papers Track. He is 21 years old outside the range of 30-50 years. Furthermore, he is a bachelor student and this is his first conference contribution.

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

## A    EXPERIMENTAL SETTINGS

For the comparison, we used the offical PyTorch (Paszke et al., 2019) implementations of Style-GAN2[1] and the GAN from Liu et al. (2021)[2]. Both GANs were trained on FFHQ datasetKarras et al. (2019). For the GANSpace, we used scikit-learn(Pedregosa et al., 2011) implementations of PCA and ICA.

For PCA, we set the number of components to 500 for both GANs. For ICA, we set the number of components to 20 and 100 for StyleGAN2 and to 500 and 1000 for the GAN from Liu et al. (2021).

## B    DETAILED ICA RESULTS

### B.1    ICA RESULTS

The effect of the number of components on ICA in GANSpace was investigated. Similar to PCA, ICA requires a number of components to be specified as a hyper-parameter. We investigated the effect of the number of components on the obtained manipulations, testing two options for Style-GAN2: 20 and 100 components (Figs. 3, 4). We found that increasing the number of components leads to a significant increase in the disentanglement of the manipulations. In addition, GANSpace with ICA discovers a more diverse set of manipulations than GANSpace with PCA, e.g. it found "weight" (Figs. 4b), "brightness" (Figs. 4e, 6b) and "background" directions which were not discovered by the original GANSpace. The FID score improved as the number of components increased. Specifically, when the manipulation strength $\alpha$ was randomly chosen from the range of $[-3, 3]$, the FID score improved from 25.48 to 20.13. Using the larger range of $[-6, 6]$, the FID score also improved from 35.16 to 25.43. These results can be seen in Table 2. This shows that with the increased number of components, the quality of transformations also increases.

For the GAN from Liu et al. (2021), ICA was applied at the $conv1$ layer as was described in Section 3. But as this layer has 16384 components, ICA converged only for 500 and 1000 components. The resulting set of manipulations can be characterised as more diverse than those obtained with PCA and similar to one produced by ICA with StyleGAN2. However, the increased number of ICA components only slightly increases the disentanglement and does not change the manipulation quality (Figs. 5, 6, Table 3).

### B.2    ICA EFFECTIVENESS AND LIMITATIONS

By substituting PCA with ICA in GANSpace, the found manipulations may become significantly more disentangled and the set of manipulations becomes more diverse (Table 2 and Fig. 4). This observation can be attributed to the fact that PCA focuses only on maximizing the variance, while ICA pays attention to the statistical independence of components. However, ICA does not always achieve disentanglement as its results highly depend on the chosen number of components. Moreover, the disentanglement of the original feature space also affects ICA results. For instance, $\mathcal{W}$ space StyleGAN2 is known as disentangled latent space (Wu et al., 2021) which makes achieving the disentanglement possible with ICA. However, the $conv1$ layer of the GAN from Liu et al. (2021) does not have underlying independent components, making ICA much less efficient than for Style-GAN2. Overall, our results suggest that ICA may help achieve disentangled image manipulations in the latent space of GANs, but its effectiveness depends on the underlying disentanglement of the feature space and the careful choice of the number of components.

We also noticed that, despite the difference in the number of parameters and dimensionality of the feature spaces, both the original GANSpace and ICA discover the same set of manipulation directions for both GANs. For example, for both StyleGAN and the GAN from Liu et al. (2021), GANSpace found "pose", "age", "skin colour" and "smile" transformations (Figs. 3, 4) and ICA discovered additional "age", "smile", "weight", "skin colour" and "brightness" manipulations (Figs. 5, 6). Thus, the latent spaces of both GANs retain similar fundamental moving directions.

---

[1]`https://github.com/NVlabs/stylegan2-ada-pytorch`
[2]`https://github.com/odegeasslbc/FastGAN-pytorch`

# C    EXAMPLES OF MANIPULATIONS AND FID SCORES

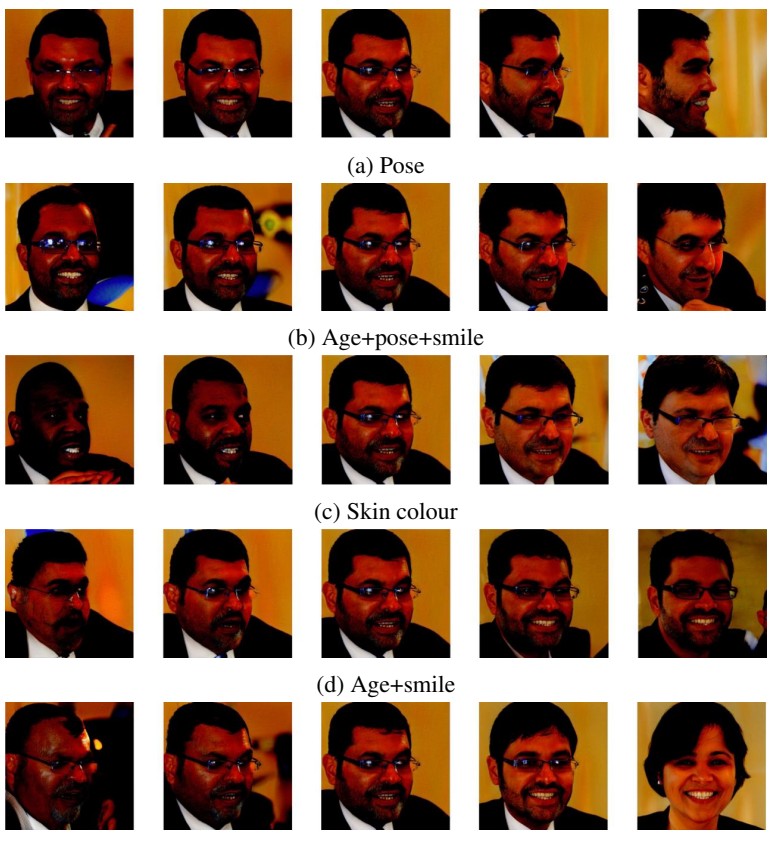

(a) Pose

(b) Age+pose+smile

(c) Skin colour

(d) Age+smile

(e) Gender+pose+age+smile

Figure 1: Sequences of image manipulations performed using directions from GANSpace applied to StyleGAN2

Table 1: FID scores for original GANSpace

| Name of GAN / Manipulation strength | StyleGAN | GAN from Liu et al. (2021) |
|---|---|---|
| $\alpha \sim U[-3, 3]$ | 25.26 | 20.41 |
| $\alpha \sim U[-6, 6]$ | 33.33 | 24.33 |

Table 2: FID scores for GANSpace for StyleGAN2 with ICA

| Number of components / Manipulation strength | 20 | 100 |
|---|---|---|
| $\alpha \sim U[-3, 3]$ | 25.48 | 20.13 |
| $\alpha \sim U[-6, 6]$ | 35.16 | 25.43 |

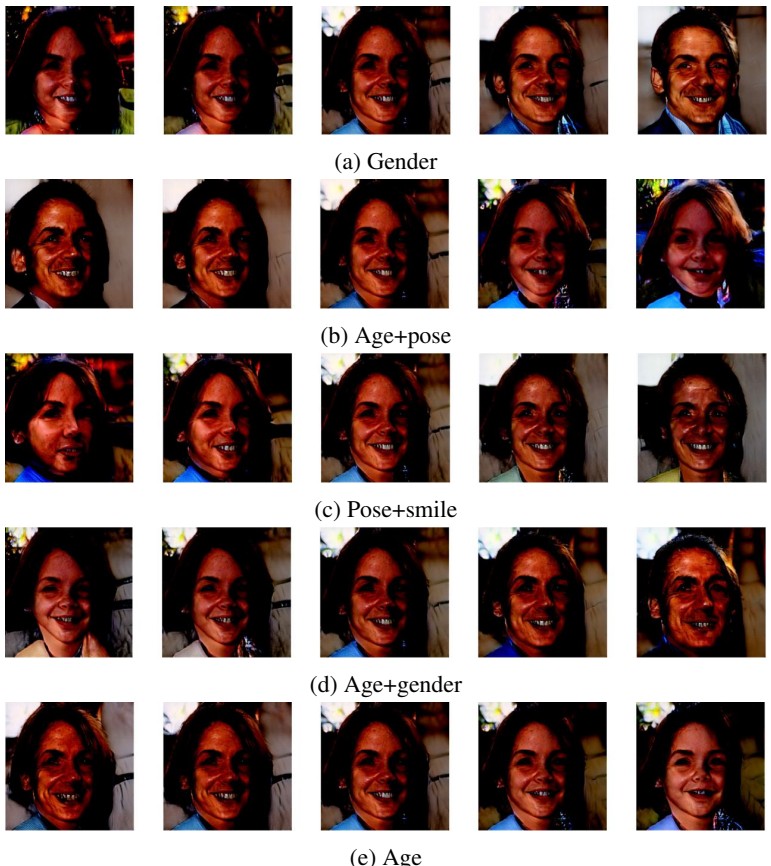

(a) Gender

(b) Age+pose

(c) Pose+smile

(d) Age+gender

(e) Age

Figure 2: Sequences of image manipulations performed using directions from GANSpace, applied to GAN from Liu et al. (2021)

Table 3: FID scores for GANSpace for GAN from Liu et al. (2021) with ICA

| Number of components / Manipulation strength | 500 | 1000 |
|---|---|---|
| $\alpha \sim U[-3, 3]$ | 20.44 | 20.92 |
| $\alpha \sim U[-6, 6]$ | 24.13 | 23.82 |

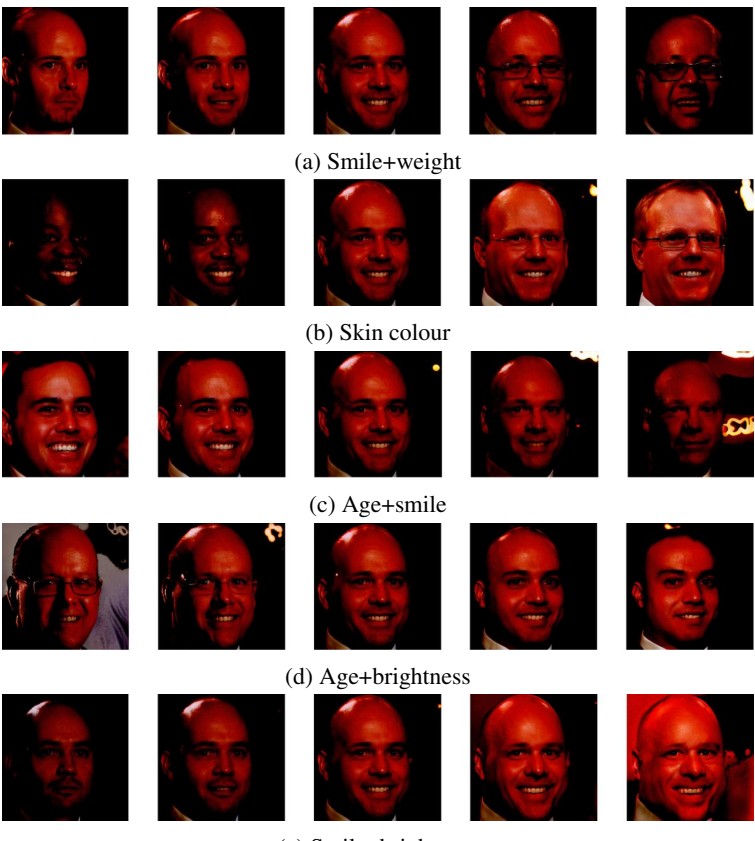

(a) Smile+weight

(b) Skin colour

(c) Age+smile

(d) Age+brightness

(e) Smile+brightness

Figure 3: Sequences of image manipulations performed using directions from GANSpace with ICA applied to StyleGAN2 (number of components is 20)

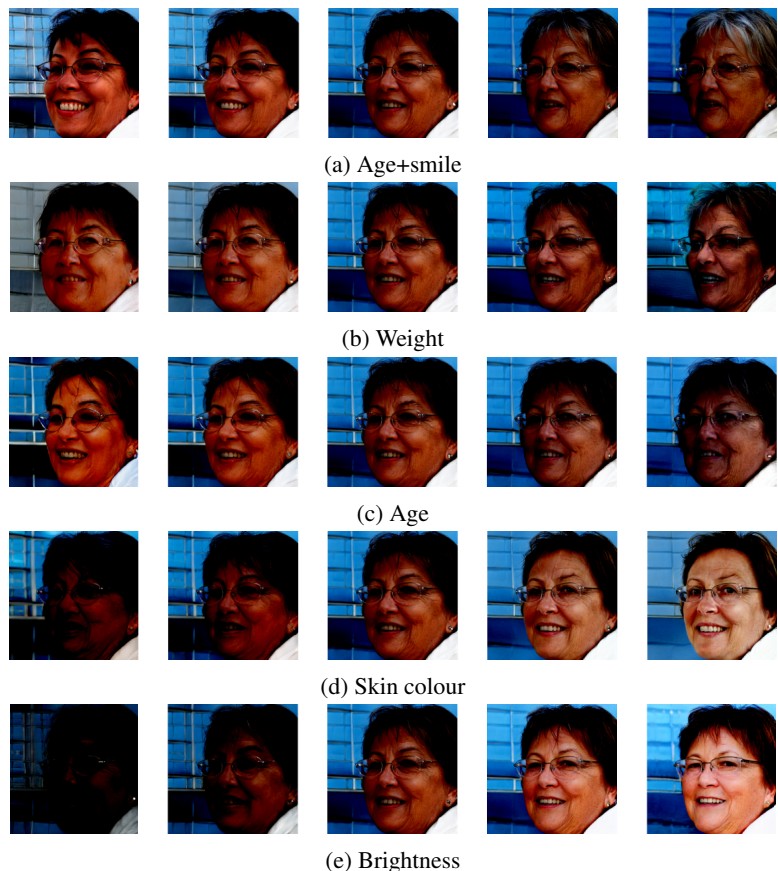

(a) Age+smile

(b) Weight

(c) Age

(d) Skin colour

(e) Brightness

Figure 4: Sequences of image manipulations performed using directions from GANSpace with ICA applied to StyleGAN2 (number of components is 100)

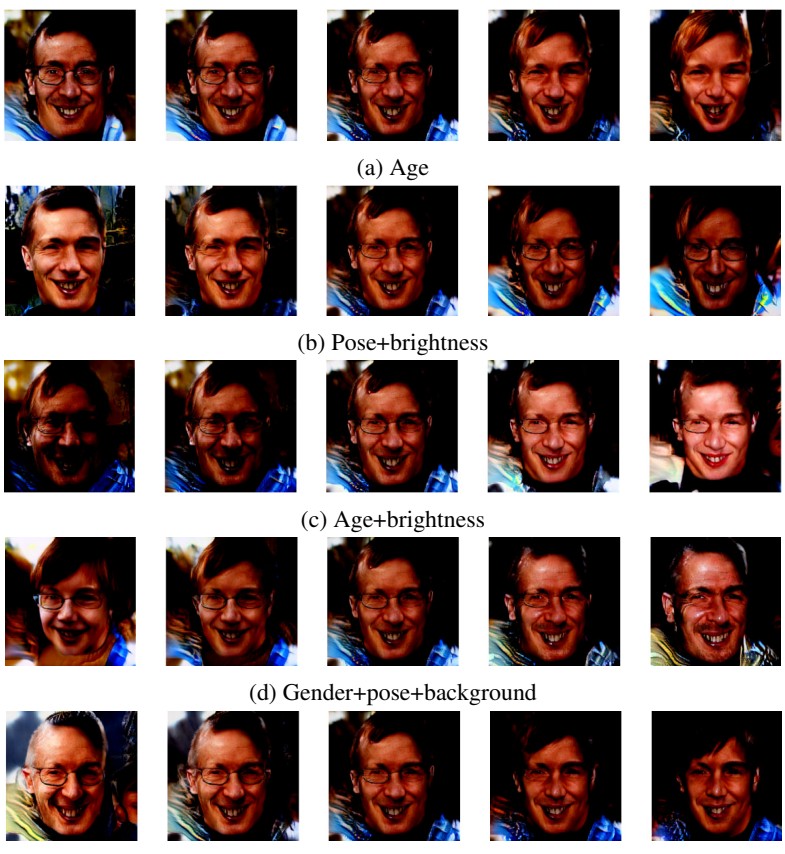

(a) Age

(b) Pose+brightness

(c) Age+brightness

(d) Gender+pose+background

(e) Age+background

Figure 5: Sequences of image manipulations performed using directions from GANSpace with ICA applied to GAN from Liu et al. (2021) (number of components is 500)

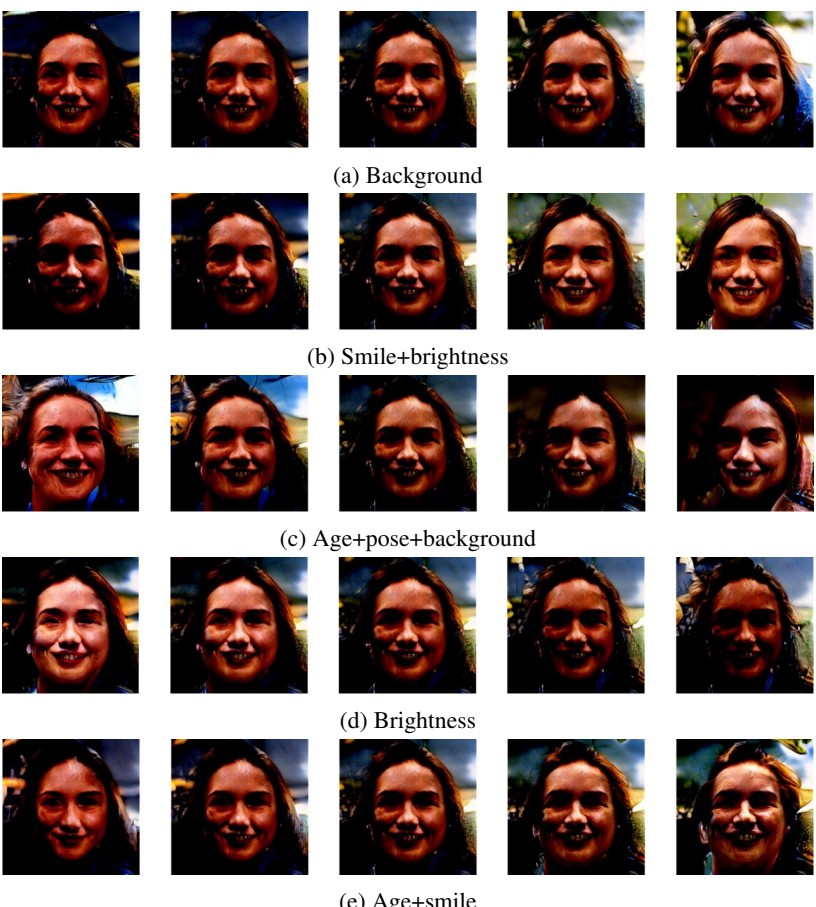

(a) Background

(b) Smile+brightness

(c) Age+pose+background

(d) Brightness

(e) Age+smile

Figure 6: Sequences of image manipulations performed using directions from GANSpace with ICA applied to GAN from Liu et al. (2021) (number of components is 1000)

