# OpenReview forum: "Exploring Semantic Variations in GAN Latent Spaces via Matrix Factorization"
_ICLR.cc/2023/TinyPapers — Submitted to Tiny Papers @ ICLR 2023_

### Official Review · Reviewer_uz6m · 2023-03-23

**Confidence:** 2

**Summary Of Contributions:**

The paper focuses on investigating image manipulation learned by GANSpace. The main idea is to use PCA and ICA on middle features of the generators then using the least squares method to obtain the matrix of transformation in the original latent space.

**Rating:**

Clear, Correct, and Reproducible (CCR): a submission which meets the reviewing criteria

**Strengths And Weaknesses:**

# Strengths
- The proposed method is new and interpretable since it is based on linear models and generalized linear models.
- The paper shows experiments on various architecture of GANs e.g., StyleGAN2.
- Qualitative results are shown to verify the proposed method.

# Weaknesses
- The paper should move the description of the method from the Appendix to the main text.

**Suggested Changes:**

- The paper should move the description of the method from the Appendix to the main text.

---

### Official Review · Reviewer_jvSY · 2023-03-31

**Confidence:** 3

**Summary Of Contributions:**

The paper explores the image manipulations learned by GANSpace, a state-of-the-art method based on PCA, and compares it to a modified version that utilizes Independent Component Analysis (ICA) to address the problem of entanglement in image manipulations. The authors demonstrate that ICA can produce more disentangled manipulations and a wider variety of transformations compared to PCA.

**Rating:**

Clear, Correct, and Reproducible (CCR): a submission which meets the reviewing criteria

**Strengths And Weaknesses:**

### Strengths:

1. Effective image manipulations. The paper demonstrates that GANSpace produces a wide range of high-quality image manipulations, making it a valuable tool for controlled data generation with GANs.

2. Replacement of PCA with ICA. The paper proposes replacing PCA with ICA, which improves the quality and disentanglement of manipulations.

3. Visual and quantitative evaluation. The paper performs both a visual and quantitative evaluation of GANSpace on two different GAN models, providing comprehensive assessment results.

### Weaknesses:

1. Limited explanation of ICA: The paper provides limited explanation of ICA and how it improves the quality and disentanglement of manipulations compared to PCA. The authors could have provided more detail on the mathematical concepts and algorithms used in ICA.

2. Writing organization. The results session is too intense to read. Authors are expected to reorganize this session into categorical explanations of their findings

**Suggested Changes:**

1. Provide a more detailed explanation of ICA and its advantages over PCA, to make it more accessible to readers.

2. Reorganize the results section. The review suggests that the results section is too intense to read and could benefit from being reorganized into categorical explanations. The reviewer could provide specific suggestions on how the authors could reorganize the section to make it more readable and accessible.

---

### Meta-Review · Area_Chair_JbmJ · 2023-04-08

**Recommendation:** Invite to archive
**Confidence:** 3

**Metareview:**

This paper explores the image manipulations learned by GANSpace and proposes a modification that utilizes Independent Component Analysis (ICA) to improve the quality and disentanglement of manipulations. The paper is well-written and provides effective image manipulations, visual and quantitative evaluation, and shows experiments on various GAN architectures. However, the paper lacks a detailed explanation of ICA and could benefit from reorganizing the results section.

**Summary:**

The paper investigates image manipulations learned by GANSpace and proposes a modification that utilizes ICA to improve the disentanglement of manipulations. The paper provides effective image manipulations, visual and quantitative evaluation, and shows experiments on various GAN architectures.

**Reason For Not Giving A Higher Recommendation:**

The paper could be improved with a more detailed explanation of ICA and its advantages over PCA to make it more accessible to readers. Reviewers agreed that the method description should be moved from the appendix to the main text. This paper could be carefully polished to ease the understanding of the audience of ICLR Tiny paper readers.

Additionally, more visual comparisons between PCA and ICA could be shown. This could improve this work to be more convincing.

To ensure the reproducibility of this work, authors are strongly encouraged to release the code used in the experiments. Experimental settings are also encouraged to present in the appendix.

**Reason For Not Giving A Lower Recommendation:**

This paper demonstrates successful image latent space manipulation using both visual results and quantitative results. The authors show results from various architectures of GANs, and the method proposed is new and interpretable since it is based on linear models and generalized linear models.

---

> ### Author Response · Authors · 2023-05-23
> **Revised Paper: Addressing Reviewer Feedback**
>
> Dear ICLR Tiny Papers Reviewers,
>
> We are writing to express our gratitude for taking the time to review our paper. Your feedback and constructive criticism have been invaluable in helping us to improve our work. We appreciate the time and effort you have put into providing detailed comments on our submission.
>
> We have carefully considered all of your suggestions and have made the following changes to the paper:
> 1. “Provide a more detailed explanation of ICA and its advantages over PCA, to make it more accessible to readers.” - we have added the “Background” section where we briefly explain PCA and ICA as well the advantages of ICA over PCA for the image manipulation in the latent space of GANs;
> 2. “Reorganize the results section. The review suggests that the results section is too intense to read and could benefit from being reorganized into categorical explanations.” - we have split the “Results and Discussion” Section into several subsections; moved the details of the “Results” in the Appendix; leaving the general summary of results in the main text;
> 3. “The paper should move the description of the method from the Appendix to the main text” - we moved the description of the method from the Appendix to the main text;
> 4. “More visual comparisons between PCA and ICA could be shown” - added more visualizations of found manipulation directions for both PCA and ICA;
> 5. “Authors are strongly encouraged to release the code used in the experiments” - pushed all the code used for the experiments to GitHub repository;
> 6. “Experimental settings are also encouraged to present in the appendix” - added the “Experimental settings” in the appendix containing the description of which GANs, PCA and ICA implementations were used in the experiments.
>
> Thank you for your time and consideration.

---

> ### Author Response · Authors · 2023-05-30
> **Opt-in for archival**
>
> In addition, we want to stay that we wish to opt-in for archival

---

### Decision · Program_Chairs · 2023-04-10

Invite to archive

---

> ### Comment · Area_Chair_JbmJ · 2023-06-06
> **Check for archive**
>
>  This work meets the threshold for archival, contents the URM statement and is deanonymized.